# An End-to-End Recurrent Neural Network for Radial MR Image Reconstruction

**DOI:** 10.3390/s22197277

**Published:** 2022-09-26

**Authors:** Changheun Oh, Jun-Young Chung, Yeji Han

**Affiliations:** 1Neuroscience Research Institute, Gachon University, Incheon 21565, Korea; 2Department of Neuroscience, College of Medicine, Gachon University, Incheon 21565, Korea; 3Department of Biomedical Engineering, Gachon University, Incheon 21936, Korea

**Keywords:** image reconstruction, RNN, end to end, MRI

## Abstract

Recent advances in deep learning have contributed greatly to the field of parallel MR imaging, where a reduced amount of *k-space* data are acquired to accelerate imaging time. In our previous work, we have proposed a deep learning method to reconstruct MR images directly from *k-space* data acquired with Cartesian trajectories. However, MRI utilizes various non-Cartesian trajectories, such as radial trajectories, with various numbers of multi-channel RF coils according to the purpose of an MRI scan. Thus, it is important for a reconstruction network to efficiently unfold aliasing artifacts due to undersampling and to combine multi-channel *k-space* data into single-channel data. In this work, a neural network named ‘ETER-net’ is utilized to reconstruct an MR image directly from *k-space* data acquired with Cartesian and non-Cartesian trajectories and multi-channel RF coils. In the proposed image reconstruction network, the domain transform network converts *k-space* data into a rough image, which is then refined in the following network to reconstruct a final image. We also analyze loss functions including adversarial and perceptual losses to improve the network performance. For experiments, we acquired *k-space* data at a 3T MRI scanner with Cartesian and radial trajectories to show the learning mechanism of the direct mapping relationship between the *k-space* and the corresponding image by the proposed network and to demonstrate the practical applications. According to our experiments, the proposed method showed satisfactory performance in reconstructing images from undersampled single- or multi-channel *k-space* data with reduced image artifacts. In conclusion, the proposed method is a deep-learning-based MR reconstruction network, which can be used as a unified solution for parallel MRI, where *k-space* data are acquired with various scanning trajectories.

## 1. Introduction

*K-space* is the sensor domain signal in magnetic resonance imaging (MRI) systems, and its trajectory is usually Cartesian. For certain purposes, such as to speed up the scanning time or to improve motion robustness, *k-space* data can be acquired with non-Cartesian trajectories [1], e.g., radial and spiral trajectories. In such cases, Radon transformation [2] or non-uniform fast Fourier transform (NUFFT) [3] can be considered for image reconstruction. Parallel imaging techniques have been used in combination with both Cartesian and non-Cartesian trajectories to reduce the data-acquisition time by undersampling. To reconstruct an image from undersampled *k-space* data in parallel imaging, many reconstruction algorithms have been developed. For example, sensitivity encoding (SENSE) [4] or generalized autocalibrating partially parallel acquisitions (GRAPPA) [5] can be adopted to reconstruct an image from regularly undersampled *k-space* data. For irregularly undersampled *k-space* data, compressed sensing [6] can be used.

As deep learning techniques are widely adopted in various imaging processing fields [7,8,9,10], a number of deep learning techniques have been introduced in the medical imaging field. Generally, deep learning techniques are adopted to reconstruct images for conventional medical imaging systems, including MRI, computed tomography (CT), single-photon emission computed tomography (SPECT) and positron emission tomography (PET), and to improve efficiency in diagnostic and research processes [11,12,13,14,15]. For MR image reconstruction, deep learning approaches can be categorized into three types: estimation of full *k-space* data from undersampled *k-space* data, transformation of a corrupted image to an artifact-free image, and reconstruction of an image from *k-space* input data. For the estimation of un-acquired data from undersampled *k-space* data, a convolutional neural network (CNN) has been developed [16,17]. Then, to reconstruct an image from the estimated full *k-space* data, a conventional transformation, such as inverse Fourier transform (IFT), can be used. To transform a corrupted input image into an artifact-free output image, multi-layer perceptrons (MLP) or CNN-based networks have been proposed [18,19,20,21]. In these cases, an initial aliased input image is obtained by IFT of the undersampled *k-space* data, and the proposed network is used to generate an anti-aliased image. To further improve the quality of an anti-aliased output image, data consistency, adversarial loss, and perceptual loss terms can be properly added, where the original *k-space* data are utilized as known prior information.

Domain transform can be performed by deep learning approaches, where an output image can be directly reconstructed from input *k-space* data [22,23]. As an analytic solution to domain transformation already exists, deep learning-based image reconstruction methods that transform *k-space* data into an image are less frequently reported than image to image or *k-space* to *k-space* reconstruction methods. The *k-space* to image domain transform in MRI requires a full receptive field, because a single point of the *k-space* data is defined by integration of the entire spatial information [24]. For the deep learning-based direct mapping method, automated transform by manifold approximation (AUTOMAP) was proposed as an end-to-end image reconstruction architecture with multiple fully connected layers [22]. In AUTOMAP, a deep learning network was used to reconstruct an image from the measured *k-space* data to correct unknown artifacts and errors. However, fully connected layers of AUTOMAP require a large memory size, which restricts image resolutions by current hardware performances.

In our previous work, we have proposed a deep learning method to reconstruct MR images directly from the k-space data acquired while solving the issues related to memory size and hardware performance. In this work, we take one step further and demonstrate that the proposed method can be used to reconstruct images acquired with various non-Cartesian trajectories, such as radial trajectories, with various numbers of multi-channel RF coils according to the purpose of the MRI scan. The novel contribution of this work is that the proposed method can be utilized as a unified solution for image reconstruction because it can be used regardless of the trajectories while overcoming the hardware limitations for computation and thereby enabling the reconstruction of higher-resolution images. The proposed network achieved this goal by replacing fully connected network with a bi-directional recurrent neural network (RNN) for end-to-end image reconstruction [25]. The core layers of the proposed network perform domain transformation through RNN-based networks, and the additional CNN-based network performs refinement for the final image. In the following sections, the proposed network will be explained and analyzed in detail.

## 2. Methods

A schematic illustration of the proposed end-to-end MR image reconstruction network (ETER-net) is presented in Figure 1. The ETER-net is an RNN-based image reconstruction technique for undersampled *k-space* data acquired with either Cartesian or non-Cartesian trajectories. The deep learning architecture in the proposed method comprises two parts: a domain transform network and a refinement network. In the domain transform network, the input *k-space* data are transformed into multiple latent features in the image domain. Then, the refinement network generates a final image from the multiple spatial features. A detailed description of the domain transformation and the refinement networks is provided in the following sections.

### 2.1. Domain Transform Network

The domain transform network should transform the *k-space* input data into an image. In a global transform, the entire *k-space* information should be reflected for the generation of a single pixel in an image. Even if some fully connected layers can learn global transformation, fully connected layers require huge parameters, which increases the amount of computation and GPU memory size. The proposed method utilizes four recurrent neural networks that sweep both horizontally and vertically in forward and reverse directions across the input *k-space* data, which reduces the demand on the number of parameters while utilizing the entire data. The domain transform network is designed to produce the output with the same width and height as the final image. Because the proposed network has different input and output sizes, it can be used to reconstruct images from *k-space* data acquired with non-Cartesian, undersampled, and arbitrary trajectories. The recurrent layers in the proposed method can be expressed as:(1)$$v_{i}^{F}=f_{FWD}\left(z_{i-1}^{F},x_{i}\right),\;\;i=1,\ldots ,I,$$
(2)$$v_{i}^{R}=f_{REV}\left(z_{i+1}^{R},x_{i}\right),\;\;i=I,\ldots ,1.$$
where $f_{FWD}$ is the forward sweeping recurrent layer and $f_{REV}$ is the reverse sweeping recurrent layer. The forward sweeping is performed from left to right in the horizontal recurrent layer and from top to bottom in the vertical recurrent layer. Similarly, the reverse sweeping is performed from right to left in the horizontal recurrent layer and from bottom to top in the vertical recurrent layer. *i* is the index in the sweeping direction, and the *i*th input vector is indicated as $x_{i}$. The output feature vectors and the hidden state vectors generated by the forward and reverse recurrent layers are represented as $v_{i}^{F}$, $v_{i}^{R}$, $z_{i}^{F}$ and $z_{i}^{R}$, respectively.

In this work, the input *k-space* data are assumed as complex-valued data with a dimension of $K_{x}\times K_{y}\times N_{c}$, where $K_{x}$ is the number of readout samples, $K_{y}$ is the number of phase encoding in the Cartesian coordinates (or the number of radial views in the radial coordinates), and $N_{c}$ is the number of Rx coils. By modifying the complex-valued data into two sets of real-valued data, input X ($\in R^{K_{x}\times K_{y}\times 2N_{c}}$) has 3D data of $K_{x}\times K_{y}\times 2N_{c}$.

Each 2D matrix of the input 3D matrix X along the $k_{x}$ direction is inserted (and flattened) into a horizontal recurrent layer, as illustrated in Figure 1. The horizontal recurrent layer sweeps the horizontal ($k_{x}$) axis in the forward and reverse bi-directions, and the length of generated outputs is $K_{x}$, which is equivalent to the input sequence length. For the horizontal recurrent layer, the number of neurons in the horizontal recurrent layer is $N_{y}\times N_{f}$, where $N_{f}$ is a hyper-parameter defined by user. Then, the input and output of the bi-directional horizontal recurrent layers can be expressed as:(3)$$v_{i}^{F,hor}=f_{FWD}\left(z_{i-1}^{F,hor},x_{i}^{col}\right),\;\;i=1,\ldots ,K_{x},$$
(4)$$v_{i}^{R,hor}=f_{REV}\left(z_{i+1}^{R,hor},x_{i}^{col}\right),\;\;i=K_{x},\ldots ,1.$$
where *i* is the index in the horizontal direction and $x_{i}^{col}$ is the flattened form of the *i*th 2D matrix of the input 3D matrix X along the $k_{x}$ direction. For the horizontal forward recurrent layer, $z_{i}^{F,hor}$ is the *i*th hidden state and $v_{i}^{F,hor}$ is the *i*th output vector. For the horizontal reverse recurrent layer, $z_{i}^{R,hor}$ is the *i*th hidden state and $v_{i}^{R,hor}$ is the *i*th output. The entire output of the horizontal recurrent layer can be represented as:(5)$$V^{hor}=\left\{\begin{array}{c} v_{1}^{F,hor},\;\ldots \;,v_{K_{x}}^{F,hor},\\ v_{1}^{R,hor},\;\ldots \;,v_{K_{x}}^{R,hor} \end{array}\right\}.$$

To transfer the output of the horizontal recurrent layer to the following vertical recurrent layer, $V^{hor}$ is reshaped as $V^{\prime }$ ( $\in R^{K_{x}\times N_{y}\times N_{f}}$ ). By vertically sweeping $V^{\prime }$ in the forward and reverse directions, the vertical recurrent layer generates an output with a sequence length $N_{y}$ and the number of neurons $N_{x}\times N_{f}$. The relationship between the input and output of the bi-directional vertical recurrent layers can be written as:(6)$$v_{j}^{F,ver}=f_{FWD}\left(z_{j-1}^{F,ver},x_{j}^{row}\right),\;\;j=1,\ldots ,N_{y},$$
(7)$$v_{j}^{R,ver}=f_{REV}\left(z_{j+1}^{R,ver},x_{j}^{row}\right),\;\;j=N_{y},\ldots ,1.$$
where *j* is the index in the vertical direction and $x_{j}^{row}$ is the flattened form of the *j*th 2D matrix of the input 3D matrix $V^{\prime }$ along the $n_{y}$ direction. For the vertical forward recurrent layer, $z_{j}^{F,ver}$ is the *j*th hidden state and $v_{j}^{F,ver}$ is the *j*th output. For the vertical reverse recurrent layer, $z_{j}^{R,ver}$ is the *j*th hidden state and and $v_{j}^{R,ver}$ is the *j*th output. Then, the entire output of the vertical RNN ( $V^{ver}$) is reshaped as $N_{x}\times N_{y}\times N_{f}$ to be used as an input for the following refinement network.

To summarize, the domain transform network generates output latent features ($\in \;R^{N_{x}\times N_{y}\times N_{f}}$) from the input *k-space* data ($\in R^{K_{x}\times K_{y}\times 2N_{c}}$).

### 2.2. Refinement Network

The latent features generated as an output of the domain transform network can be regarded as a roughly reconstructed image. To merge and refine the latent features into a final image, the refinement network should perform a simple image-domain to image-domain task. Thus, various types of neural network architectures can be used to generate a single channel magnitude image from multiple channel spatial features, where the widths and heights of the spatial features and the final image are equivalent. We evaluated CNN-based networks for the refinement network because CNN can provide the best performance in terms of image quality. To find an optimal network for the ETER-net, we compared the single convolutional layer, the multiple convolutional layers inspired by winner-take-all autoencoder (WTA) [26], and a dual frame U-net (DFU) [27], which is one of the most commonly used architectures.

### 2.3. Loss Function

The most frequently used loss function, particularly for regression tasks, is the Euclidean distance between the given value of the label and the output of the network. However, using only the Euclidean distance as a loss function tends to blur the final image [28,29,30,31]. In order to overcome this problem, additional losses have been examined, including the adversarial loss, which uses a discriminator network to classify real and generated data [32], and the perceptual loss, which utilizes the distance of feature levels between the real and generated data examined by a pre-trained network [33]. To optimize the performance of the proposed reconstruction network, we utilize Euclidean distance, the adversarial loss based on the patch generative adversarial network (GAN) discriminator [29], and perceptual loss based on the pretrained "VGG" network [34]. The total loss function can be written as:(8)$$L_{total}=\lambda_{L1}L_{L1}+\lambda_{adv}L_{adv}+\lambda_{VGG}L_{VGG}.$$

## 3. Experiments

### 3.1. Data Preparation

As explained, the proposed network is applicable for the reconstruction of images from *k-space* data acquired with Cartesian or non-Cartesian trajectories. To demonstrate image reconstruction for various *k-space* trajectories, both Cartesian and non-Cartesian *k-space* data were acquired using FSE at a 3T MRI (Siemens Verio) with the following parameters: For Cartesian *k-space* data, FOV = 220 × 220 mm${}^{2}$, matrix size = 216 (FE) × 216 (PE), slice thickness = 5 mm, TR/TE = 5000/90 ms, and echo train length = 18. For radial *k-space* data, FOV = 256 × 256 mm${}^{2}$, matrix size = 256 (frequency encoding) × 400 (view angles), slice thickness = 5 mm, TR/TE = 5000/52 ms, and echo train length = 8.

Then, for MRI in Cartesian coordinates, the label images were prepared by Fourier transform of the full *k-space* data of each receive coil channel, taking the root mean square of the multi-channel images. For radial MRI, images from each receive coil channel were reconstructed by the filtered back projection, and the root mean square of the multi-channel images was calculated to generate the label images. For stabilization of the learning process, the *k-space* values and the intensity values of the label images were normalized to [−1,1] and [0,1], respectively.

From the acquired *k-space* data, accelerated parallel acquisition was simulated by retrospective sub-sampling of *k-space* data along the phase encoding direction for Cartesian *k-space* data or along the view angles for radial *k-space* data with various reduction factors. For Cartesian MRI, both regular and irregular undersampling patterns were used, as illustrated in Figure 2a. The number of channels of the receive coil was eight. A total of 1520 images from 14 subjects were used for training, and 64 images from four subjects were used for testing. For radial MRI, 2538 images from 26 subjects were used for training and 48 images from four subjects were used for testing. The number of channels of the receive coil for radial MRI was four. To verify the performance with respect to various acceleration factors, we examined the proposed method with reduction factors R of 2, 4, 6 and 8, respectively.

Accelerated MRI usually utilizes multi-channel receive coils to incorporate the local sensitivity of each channel of each receive coil for image reconstruction. To verify the reconstruction performance of the proposed method in case of accelerated imaging with a single-channel receive coil, we attempted to reconstruct images from a single-channel *k-space* data. For this experiment, the label image was reconstructed from fully acquired single-channel *k-space* data, and the input *k-space* data of the deep network were undersampled with a reduction factor of four.

### 3.2. Experimental Settings

From the acquired *k-space* datasets, images were reconstructed by the proposed ETER-net, two image-domain to image-domain reconstruction methods (dual frame U-net [27] and DAGAN [19]), and a k-domain to k-domain reconstruction method (*k-space* deep learning [16]). In the domain transformation network of the ETER-net, the user-defined parameter $N_{f}$ was 32, which was the maximum possible value, and it was limited by the memory size of the GPU we used. If $N_{f}$ increases as the model size becomes larger, the loss values of the network converge at lower values. As illustrated in Figure 3, we find that the proposed model with a small $N_{f}$ is not enough for end-to-end image reconstruction. In this work, we set the $N_{f}$ as the maximum possible value to maximize the size of the proposed model.

For the refinement network, experiments were performed with the following settings. In case of the single convolutional layer, it was designed to merge its input features with $N_{f}$ = 32 channels into a single channel output. For the multiple convolutional layers (WTA), which were also used in AUTOMAP [22,26], the output channels of convolutional layers were set as 256 and the final deconvolutional layer was designed to merge the features of 256 channels into a single channel output. While the vanilla U-net only utilized a skip connection, the dual frame U-net (DFU) utilized an additional by-pass connection for residual learning [27] in combination with the skip connection, resulting in improved learning performance.

Various loss functions were analyzed, where the adversarial loss and the perceptual loss were added to the ETER-net, while the pixel-wise loss function with a weighting factor (λ) of 1 was utilized as a baseline. For the perceptual loss, the feature-level difference was used as a loss based on the pre-trained VGG network. The loss functions utilizing either adversarial loss, perceptual loss, or both were compared to evaluate the effectiveness of the loss functions. Other parameters are described as follows. For Cartesian MRI, λ for adversarial loss was also set as 1 when only the adversarial loss was additionally used in combination with the baseline loss function. If only the perceptual loss was added to the baseline loss function, λ = 50 was used for perceptual loss. When both adversarial loss and perceptual loss were additionally used, λ = 0.02 and λ = 5 were used for adversarial loss and perceptual loss, respectively. Then, the total number of parameters of the ETER-net utilizing the DFU for the refinement network is 312,367,393. For radial MRI, λ = 0.001 was used for adversarial loss where the adversarial loss was additionally used in combination with the baseline loss function. When only the perceptual loss was added to the baseline loss function, λ = 0.01 was used for perceptual loss. When both adversarial loss and perceptual loss were additionally used, λ = 0.001 and λ = 1 were used for adversarial loss and perceptual loss, respectively. Then, the total number of parameters of the ETER-net was 453,460,769. The training was commonly performed as in Adam [35] with lr = 0.0001.

The proposed architecture was implemented on Pytorch [36] and computed by an Intel(R) Core (TM) i5-8400 2.80GHz CPU and NVIDIA TITAN XP GPU. We used the gated recurrent unit (GRU) [37] for recurrent units to reduce the number of parameters. There was no significant difference in the performance compared with the long short-term memory (LSTM) [38].

### 3.3. Quantitative Evaluation

The normalized mean square error (nMSE) and structural similarity index (SSIM) [39] are deployed for the quantitative evaluation. They are defined as follows:(9)$$nMSE=\frac{\sum_{i}^{I}\sum_{j}^{J}{\left[x_{i,j}^{GT}-x_{i,j}^{PRED}\right]}^{2}}{\sum_{i}^{I}\sum_{j}^{J}{\left[x_{i,j}^{GT}\right]}^{2}},$$
(10)$$SSIM=\frac{\left(2\mu_{x}\mu_{y}+c_{1}\right)\left(2\sigma_{xy}+c_{2}\right)}{\left(\mu_{x}^{2}+\mu_{y}^{2}+c_{1}\right)\left(\sigma_{x}^{2}+\sigma_{y}^{2}+c_{2}\right)}.$$
where $x_{i,j}^{GT}$ is the ground truth image, $x_{i,j}^{PRED}$ is the predicted image, $\mu_{x}$ and $\mu_{y}$ are the means of *x* and *y*, respectively, $\sigma_{x}^{2}$ is the variances of *x*, $\sigma_{y}^{2}$ is the variances of *y*, and $\sigma_{xy}$ is the covariance of *x* and *y*. To prevent divergence by dividing by zero, small constants are included such as $c_{1}$ and $c_{2}$.

## 4. Results

### 4.1. Variants of the Refinement Network

Figure 4 presents the reconstructed images from three different refinement networks: the single convolution layer, multiple convolutional layers, and dual frame U-net. Figure 5 presents the quantitative analysis of the three refinement networks. As shown in Figure 4, the images refined by the single convolution layer contain a substantial amount of structural artifacts. In addition, the image details are also deteriorated. In contrast, the images refined by the WTA and DFU have considerably improved image quality, and the image details are clear. However, the images refined by the DFU have better image details. The quantitative measures also demonstrate that the DFU can be efficiently used for the refinement of the latent features in the proposed method. The single convolutional layer shows considerably higher *nMSE* and lower *SSIM* values than the other two methods, and the DFU shows improved performance in terms of *nMSE* and *SSIM*.

### 4.2. Acceleration with a Single-Channel RF Coil

While most parallel MRI methods utilize information from multi-channel RF coils to reconstruct images from undersampled *k-space* data, deep learning-based reconstruction approaches can generate unaliased images from an undersampled dataset without using the information of the multi-channel receive coil [40]. To demonstrate the performance of the proposed ETER-net, we performed image reconstruction from the undersampled *k-space* data obtained with a single receive channel. To compare the results with a commonly used deep learning reconstruction method, we also simulated the image-domain to image-domain reconstruction using the dual frame U-net. In this experiment, the label images were reconstructed from fully sampled *k-space* data of a single receive channel. Figure 6 shows three different slices reconstructed from a single channel *k-space* data, which were undersampled with a reduction factor of four. All images show non-uniform intensities because the *k-space* data are acquired from a single receive channel, which has non-uniform sensitivity. The images reconstructed by the DFU (column (b)) and the ETER-net (column (d)) are very similar to the label images in column (a). However, when the images are brightened to emphasize artifacts (columns (c) and (e)), it is clear that the proposed ETER-net generates images with less artifacts than the DFU. In particular, the images reconstructed by the DFU have more blurred structural edges and apparent aliasing artifacts, while the ETER-net generates alias-free images with improved image details.

### 4.3. Loss Functions

To further optimize the proposed network for image reconstruction, two different losses were additionally calculated along with the baseline loss function of the Euclidean distance: adversarial loss and perceptual loss. The effects of the loss functions were analyzed for the undersampled *k-space* data with R = 4. The ETER-net with different loss functions was also compared with two different image-domain to image-domain reconstruction methods, DFU [27] and DAGAN [19], and a k-domain to k-domain reconstruction method, *k-space* deep learning for accelerated MRI [16]. Figure 7 shows the result images and errors, and Figure 8 shows the quantitative analysis of the reconstructed images. Figure 7b,c show a substantial amount of image artifacts and aliasing patterns, owing to the regular subsampling of the input *k-space* data. In contrast, the proposed methods show quality results, and the aliasing artifacts are eliminated in these images (columns (e–h)). By comparing the images reconstructed with various loss functions, it can be concluded that the loss functions, including the Euclidean distance, adversarial loss, and perceptual loss, provided the best performance in terms of image quality.

### 4.4. Radial Sampling Data

The proposed method can be applied to the data acquired with a non-Cartesian trajectory because it is a unified solution for image reconstruction. As an example of non-Cartesian trajectories, we examined the proposed method for *k-space* data obtained with a radial trajectory. For the reconstruction of images from the radially under-sampled *k-space* data with a reduction factor of four, we used the ETER-net with the DFU as the refinement network. The effectiveness of various loss functions was also analyzed for the reconstruction of radially sampled *k-space* data by comparing the baseline loss function of the Euclidean distance with the additional adversarial and perceptual losses.

The reconstructed images from radially sampled *k-space* data are shown in Figure 9. When the filtered back projection was used to reconstruct the images from the undersampled radial *k-space* data, the streaking artifacts could be observed in the images, as shown in Figure 9 (rows 2). In contrast, the ETER-net provided improved image quality (rows 3 and 4). Moreover, the comparison of the images in rows 3 and 4 demonstrates that the loss function including Euclidean distance, adversarial loss, and perceptual loss improved the image quality. The improvement owing to the additional loss functions can also be validated by the quantitative measures in Table 1. The average *nMSE* is 3.51% for the filtered back projection, 1.98% for the proposed method with L1 loss only, and 1.49% for the proposed method with additional losses. In terms of *SSIM*, the average score of the filtered back projection is 0.786, the average score of the proposed method with L1 loss only is 0.922, and the average score of the proposed method with additional losses is 0.938. The proposed method with additional losses shows the least error and highest *SSIM*.

## 5. Discussion

In this study, a neural network architecture that reconstructs an MR image from *k-space* data was proposed by utilizing the RNNs. The proposed method can be used to reconstruct an image from *k-space* data acquired with various *k-space* trajectories, including Cartesian, non-Cartesian, fully sampled and under-sampled datasets, because the proposed network is a unified solution for image reconstruction. Moreover, it is applicable both for *k-space* data acquired with single or multi-channel RF coils.

Likewise, AUTOMAP can learn a direct sensor-to-image mapping. However, AUTOMAP has a major drawback, i.e., the limitation of the size of the reconstructed image, owing to the hardware performance. Because of the fully connected layers, AUTOMAP requires the number of parameters as $O\left(n^{2}\right)$ to produce an n×n image. However, the proposed method requires considerably fewer parameters by utilizing recurrent neurons, thus enabling the reconstruction of high-resolution images. To produce an n×n image, the proposed method requires the number of parameters as $O\left(n\right)$. Thus, the main advantage of the proposed method in comparison with the AUTOMAP is a considerable reduction in the number of parameters for the network.

The proposed method is a deep learning architecture that generates a final image from the input *k-space* data; this implies that it requires a domain transform to the image domain. We assumed that bi-directional recurrent layers can be used to perform the domain transform. To verify this, we examined the output features of the recurrent layers. The domain transform network of the proposed method successfully conducts domain transformation from sensor-to-image.

For refinement network, experiments were performed by applying three different CNNs. The results showed that there was a significant difference in the performance depending on which network was used for the refinement network. The performance was better when the DFU was applied, which is an advanced CNN-based network, compared with when the single convolutional layer was used. A comparable performance was shown when multiple convolutional layers were used.

The proposed method was applied to the radial trajectory as an example of non-Cartesian trajectory. For the experiments on radial trajectory, the label was defined as the reconstructed image from the Radon transform using 400 views. As demonstrated by the experiments, the view and readout axes in the radial trajectory correspond to the phase encoding and frequency encoding axes in the Cartesian trajectories, respectively, and the image reconstruction using the proposed ETER-net was successfully performed.

In general, the proposed method also provided blurry results owing to the regression problem in which the Euclidean distance was used as a loss function. To address this, learning experiments were conducted by including adversarial loss and perceptual loss. The adversarial loss encouraged the network to favor realistic solutions by trying to fool the discriminator network. Furthermore, perceptual loss allowed the network to reconstruct fine details and edges. Through these experiments, blurry results were obtained again when only the Euclidean distance was used as the loss function, and it was verified that this could be overcome by using additional losses.

## 6. Conclusions

In this work, we have proposed an efficient alternative for a deep learning-based direct mapping method that reconstructs images from undersampled *k-space* data. While previous direct mapping methods could be selectively used in cases of relatively low-resolution images due to limitations, we have demonstrated that the proposed method can be used for higher-resolution images by utilizing the inductive bias that comes from the structural characteristics of bi-RNN, where shared weights are used for sequentially inserted inputs, and information from the past can be integrated through the hidden state. The proposed method can be used as a unified solution applicable to various scanning trajectories, such as radial, spiral, and Cartesian trajectories, and it can be applied to single and multi-channel data. To verify the proposed method, experiments were conducted with an actual Cartesian and radial dataset, and both quantitative and qualitative results showed that the proposed method is effective for MR image reconstruction.

## Figures and Tables

**Figure 1 sensors-22-07277-f001:**
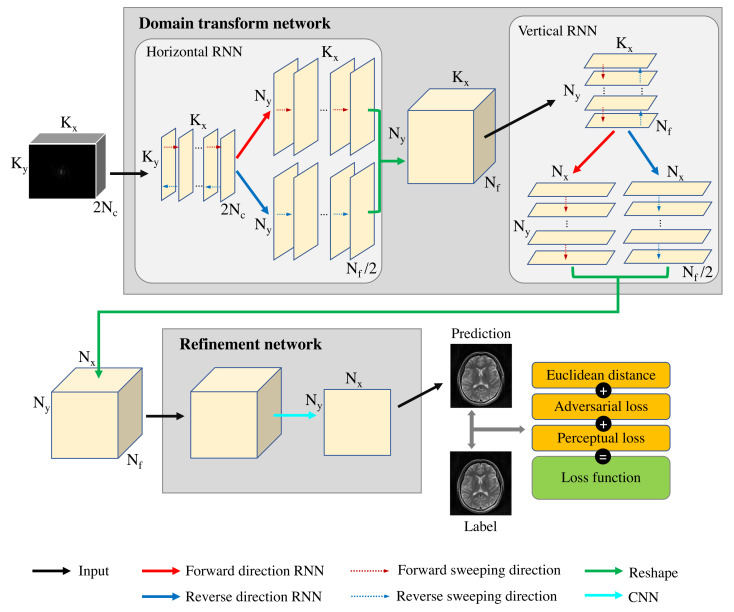
Proposed network architecture.

**Figure 2 sensors-22-07277-f002:**
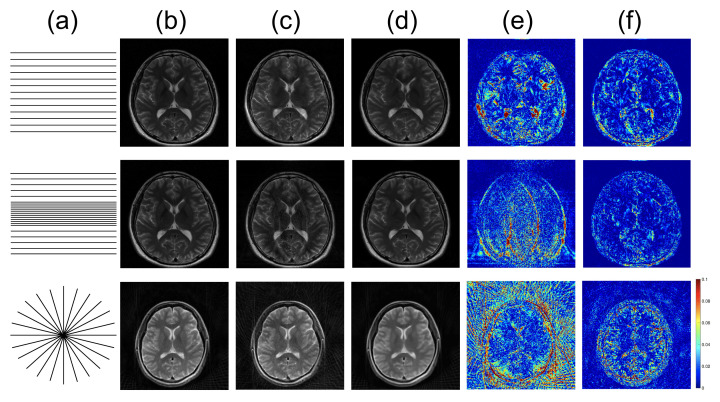
Reconstructed images using conventional and proposed methods for various trajectories: (**a**): scanning trajectory, (**b**): label image, (**c**,**e**): images reconstructed by conventional method: DAGAN (**first row**), GRAPPA (**second row**), and inverse Radon transform (**third row**), (**d**,**f**): images reconstructed by the proposed method. (**e**,**f**): Corresponding error maps in the 1/10 range.

**Figure 3 sensors-22-07277-f003:**
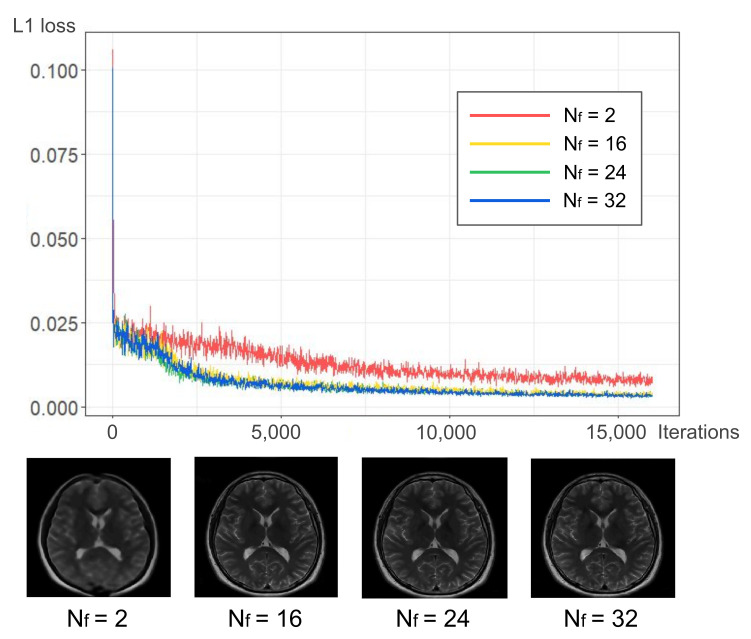
Learning curve of the proposed network corresponding to four different channel sizes: $N_{f}$ = 2, 16, 24, and 32. The reconstructed images are shown for different $N_{f}$.

**Figure 4 sensors-22-07277-f004:**
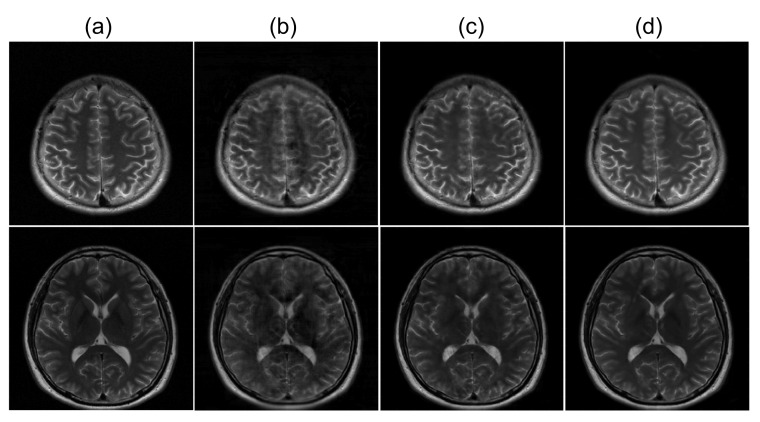
Comparison of three different refinement networks (Section 4.1). Two example slices of experiment results are shown. (**a**): Ground truth magnitude images (label images). (**b**–**d**): Reconstructed images using (**b**) single convolutional layer, (**c**) WTA, and (**d**) DFU.

**Figure 5 sensors-22-07277-f005:**
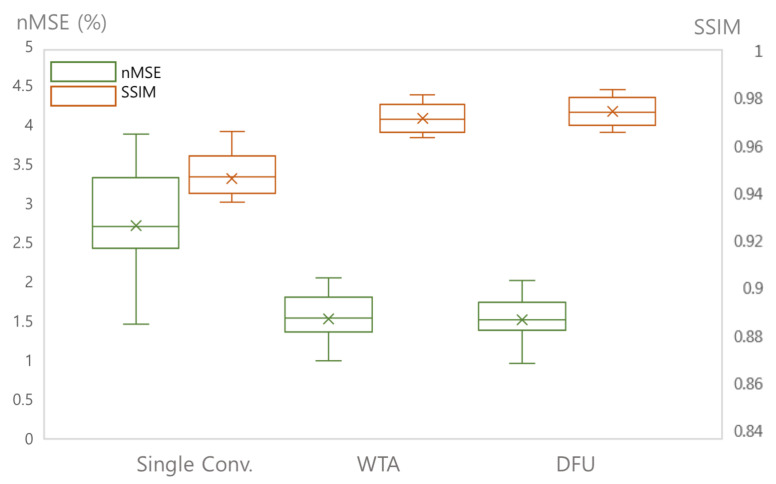
Quantitative analysis of three different refinement networks: single convolutional layer (Single Conv.), multiple convolutional layers with a deconvolutional layer (WTA), and DFU.

**Figure 6 sensors-22-07277-f006:**
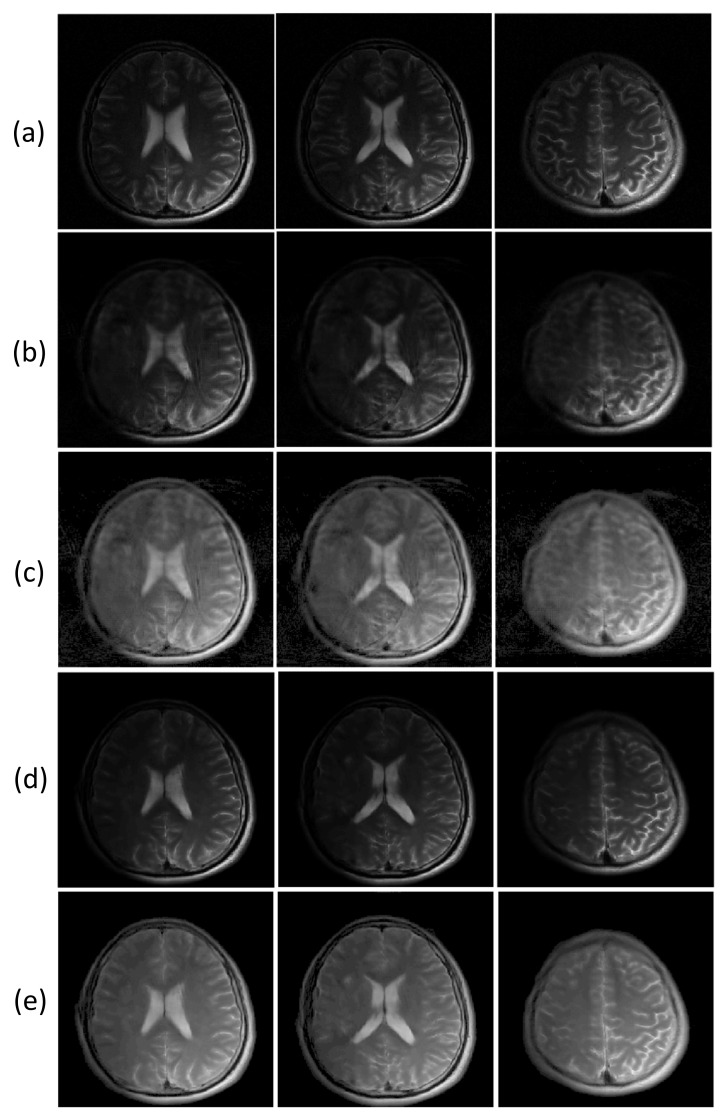
Images reconstructed from a single channel k-space data, which are undersampled with a reduction factor of four: (**a**): label images, (**b**,**c**): images reconstructed by the DFU and the corresponding brightened images, (**d**,**e**): images reconstructed by the ETER-net and the corresponding brightened images. Brightening was performed by taking the root of each value in the image.

**Figure 7 sensors-22-07277-f007:**
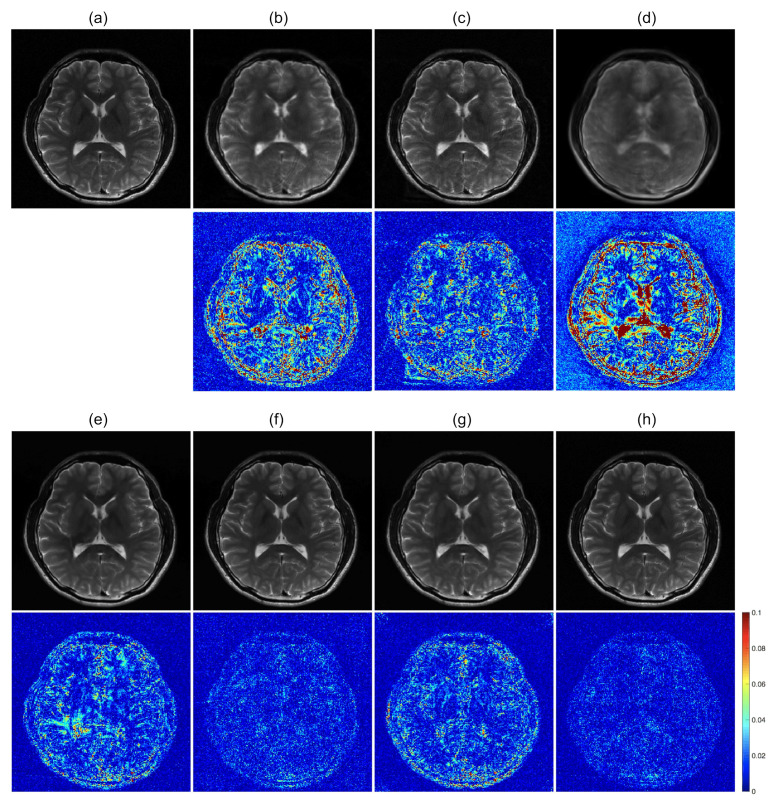
Comparison of previous methods with the proposed method. **first and third row**: Magnitude images. **second and fourth row**: Corresponding error maps in the 1/10 range. (**a**): Ground truth magnitude images (label images). The images are reconstructed by (**b**): DFU, (**c**): DAGAN, (**d**): k-space deep learning, (**e**): baseline loss, (**f**): baseline loss + perceptual loss, (**g**): baseline loss + adversarial loss, and (**h**): baseline loss + adversarial loss + perceptual loss.

**Figure 8 sensors-22-07277-f008:**
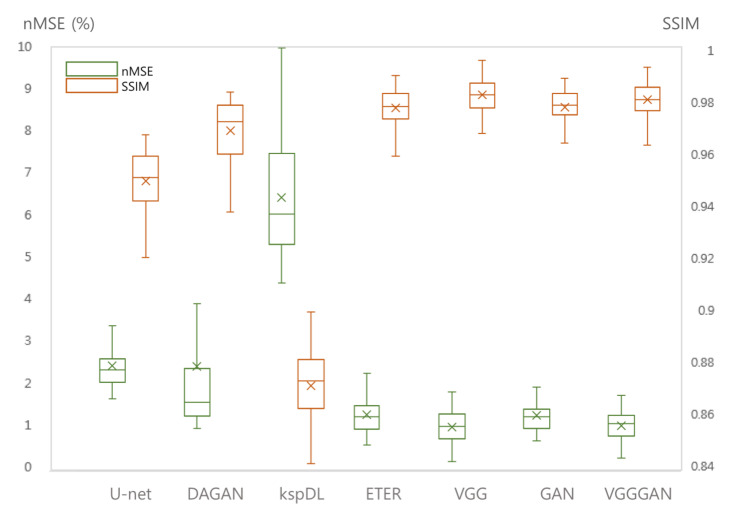
Quantitative analysis of DFU, DAGAN, and the proposed methods with/without additional losses. nMSE and SSIM were measured from 64 slices of four subjects. U-net, DAGAN, kspDL: comparison methods, ETER: with baseline loss function of pixel-wise Euclidean distance, GAN: with additional adversarial loss function, VGG: with additional perceptual loss via pre-trained VGG network, GAN + VGG: with additional adversarial and perceptual loss functions.

**Figure 9 sensors-22-07277-f009:**
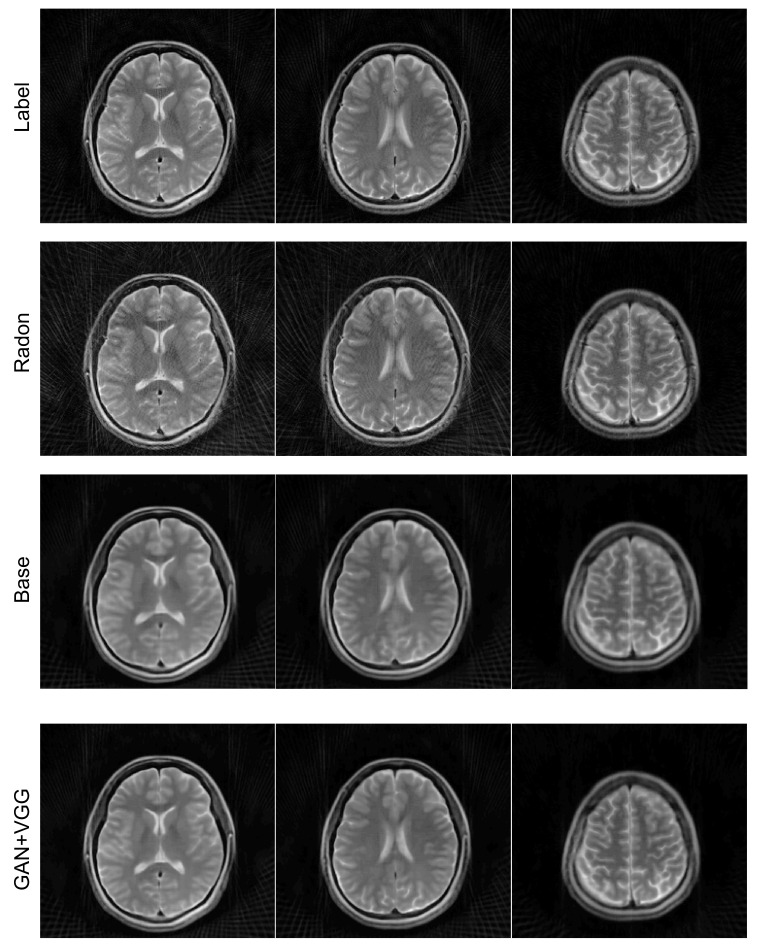
Comparison of inverse Radon transform with the proposed method with/without additional losses. Four example images reconstructed from radial *k-space* data with 100 view angles are shown. Label: reconstructed images from radial *k-space* data with 400 view angles by inverse Radon transform, Radon: inverse Radon transform, Base: the proposed method with the baseline loss function, GAN + VGG: the proposed method with additional adversarial and perceptual losses.

**Table 1 sensors-22-07277-t001:** Statistics for the proposed and the inverse Radon transform from 100 view angles of radial k-space data. Test set is 48 images from four subjects.

Case	nMSE (%)	SSIM
Radon	3.51 ± 0.63	0.786 ± 0.019
Baseline (Euclidean distance)	1.98 ± 0.40	0.922 ± 0.013
Baseline + Adversarial loss + Perceptual loss (Euclidean distance)	1.49 ± 0.45	0.938 ± 0.015

## Data Availability

Not applicable.

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
