# Peer review of "An End-to-End Recurrent Neural Network for Radial MR Image Reconstruction"

_sensors, 2022, doi:10.3390/s22197277_

Round 1

Reviewer 1 Report

It is a well-structured paper with interesting results. However, it requires further improvements before publication. (1) In the abstract, the author should highlight the specific problems to be solved in this study at the beginning, and then lead to the solutions. At present, the description is not clear. At the end of the abstract, the author can briefly summarize the research conclusions. The current expression is too redundant and should be deleted appropriately. (2) In the introduction section, you should give the novelty and the contributions of your works. (3) Proofread the paper carefully to improve it grammatically. (4) Very brief literature is presented, try to update it with some latest references. The following may be considered. [1]https://doi.org/10.3390/agriculture12060793; [2]https://doi.org/10.1109/JSTARS.2021.3059451; [3]https://doi.org/10.1109/TR.2022.3180273; [4]https://doi.org/10.1016/j.asoc.2022.109419 (5) The theoretical background of the proposed method is adequately detailed in the paper. (6) In Section 6, the conclusion and motivation of the work should be added in a more clear way. (7) At Line 141 and 142, GAN, VGG, and so on, these abbreviations need to be written in full for the first time. (8) At Line 62, about fully-sampled and under-sampled data, how to distinguish fully sampled and under sampled data. Please give the describing about them. (9) In section 2.3, the authors should give the expression of Loss function. (10) In Section 3.2, the values of parameters could be a complicated problem itself, how the authors give the values of parameters? (11) What is the motivation of the work? The problem considered does not have a sound motivation. The authors should clearly demonstrate the scientific interest of the objectives and results. (12) At Line 337, add the sections of the “Institutional Review Board Statement”, “Informed Consent Statement”, “Data Availability Statement”. (13) Correct typological mistakes and mathematical errors.

Reviewer 2 Report

See attached pdf.

Round 2

Reviewer 1 Report

I have appreciated the deep revision of the contents and the present form of this manuscript. All my previous concerns have been accurately addressed. I think that this paper can be accepted.